# Effect of Pemafibrate on Hemorheology in Patients with Hypertriglyceridemia and Aggravated Blood Fluidity Associated with Type 2 Diabetes or Metabolic Syndrome

**DOI:** 10.3390/jcm12041481

**Published:** 2023-02-13

**Authors:** Tomohiro Iwakura, Takanori Yasu, Takashi Tomoe, Asuka Ueno, Takushi Sugiyama, Naoyuki Otani, Shinya Kawamoto, Hiroyuki Nakajima

**Affiliations:** 1Department of Cardiovascular Medicine and Nephrology, Dokkyo Medical University, Nikko Medical Center, 145-1 Moritomo, Nikko 321-2593, Tochigi, Japan; 2Department of Cardiovascular Surgery, Yamanashi University, 1110 Shimokato, Chuo-shi 409-3839, Yamanashi, Japan; 3Department of Cardiovascular Surgery, Sakakibara Heart Institute, 3-16-1 Asahi-cho, Fuchu-shi 183-0003, Tokyo, Japan

**Keywords:** diabetes mellitus, free fatty acid, metabolic syndrome, microcirculation, rheology, triglyceride

## Abstract

Persistent high serum triglyceride (TG) and free fatty acid (FFA) levels, which are common in metabolic syndrome and type 2 diabetes, are risk factors for cardiovascular events because of exacerbated hemorheology. To explore the effects of pemafibrate, a selective peroxisome proliferator-activated receptor alpha modulator, on hemorheology, we performed a single-center, nonrandomized, controlled study in patients with type 2 diabetes (HbA1c 6–10%) or metabolic syndrome, with fasting TG levels of ≥ 150 mg/dL and a whole blood transit time of > 45 s on a microarray channel flow analyzer (MCFAN). Patients were divided into a study group, receiving 0.2 mg/day of pemafibrate (*n* = 50) for 16 weeks, and a non-pemafibrate control group (*n* = 46). Blood samples were drawn 8 and 16 weeks after entry to the study to evaluate whole blood transit time as a hemorheological parameter, leukocyte activity by MCFAN, and serum FFA levels. No serious adverse events were observed in either of the groups. After 16 weeks, the pemafibrate group showed a 38.6% reduction in triglycerides and a 50.7% reduction in remnant lipoproteins. Pemafibrate treatment did not significantly improve whole blood rheology or leukocyte activity in patients with type 2 diabetes mellitus or metabolic syndrome complicated by hypertriglyceridemia and exacerbated hemorheology.

## 1. Introduction

Persistently high serum triglyceride (TG) and free fatty acid (FFA) levels, which are common in type 2 diabetes and metabolic syndrome, are residual risk factors after significant low-density lipoprotein cholesterol (LDL-C) reduction by statin therapy for atherosclerotic coronary vascular disease (ASCVD) [1,2,3]. With the recent increase in the incidence of diabetes mellitus, obesity, and dyslipidemia, and with mortality from ASCVD projected to exceed 23 million per year by 2030, the appropriate management of hypertriglyceridemia and FFAs is critical for the prevention of ASCVD [4]. However, there are no definitive data showing that reducing TG reduces cardiovascular events, and the impact of TG- and FFA-lowering therapy on microcirculation is still unknown [5].

Clinical trials with niacin [6] and fenofibrate [7,8] failed to show conclusive cardiovascular outcome data indicating that lowering serum TG levels may reduce cardiovascular events. However, it has been suggested that patients with low high-density lipoprotein cholesterol (HDL-C) may clinically benefit from a lowering of TG and FFA levels, especially if they have concomitant type 2 diabetes [9,10,11]. Omega-3 fatty acids decrease TG levels and have favorable effects on inflammatory, oxidative, and thrombotic factors. [12] However, the results of the two most recent clinical trials on high-volume n-3 fatty acids in patients with elevated TG levels, despite the use of statins, are controversial regarding cardiovascular outcomes [13,14].

Previous studies showed that elevated plasma FFA and serum TG levels during intravenous lipid/heparin infusion resulted in endothelial and microvascular dysfunction in healthy subjects [15,16,17,18]. Pemafibrate, a potent and superior peroxisome proliferator-activated receptor alpha (PPARα) modulator, drastically reduces serum TG levels and improves other lipid levels. Pemafibrate ameliorates diabetic nephropathy in db/db mice via the inhibition of renal lipid content and oxidative stress [19]. To test our hypothesis that significant improvements in TG, remnant-like particle cholesterol (RLP-C), and HDL-C levels with pemafibrate treatment normalize hemorheology, we conducted a single-center, nonrandomized, controlled trial to evaluate the effects of pemafibrate treatment on hemorheology and serum FFA levels.

## 2. Materials and Methods

### 2.1. Patients

The Ethics Committee of the Dokkyo Medical University Nikko Medical Center approved the study protocol. (Nikko 31016, 31017). The study design was a single-center, non-randomized, controlled study. The inclusion criteria were as follows: (1) aged ≥20 years; (2) type 2 diabetes mellitus (HbA1c 6–10%) or metabolic syndrome [20], comorbid with fasting serum TG levels of ≥150 mg/dL; and (3) a whole blood transit time (corrected) of >45 s on microarray channel flow analyzer (MCFAN). The exclusion criteria were as follows: (1) patients treated with pemafibrate, fibrates, or omega-3 fatty acids; (2) patients deemed by the principal investigator or sub-investigator to be inappropriate for participation in the study, such as those with poor medication adherence, dementia (Hasegawa’s dementia scale <20/30), or alcoholism. Finally, 96 patients were enrolled from May 2020 until December 2021, and 50 patients were newly prescribed with 0.2 mg/day of pemafibrate for 16 weeks, and 46 patients were not prescribed with pemafibrate according to the attending physicians’ decision. Blood samples were collected at baseline, and 8 weeks (±2 weeks) and 16 weeks (±3 weeks) after the start of the treatment in the same manner. All participants provided written informed consent.

### 2.2. Assessment of Whole Blood Rheology and Leukocyte Activity Using an Ex Vivo Microchannel Model

We used a microchannel flow analyzer as an ex vivo model of capillaries and arterioles to assess whole blood rheology and leukocyte activity as previously described [16,17,21,22]. Briefly, microgrooves that formed on the surface of a silicon chip were converted to leak-proof microchannels by being tightly covered with an optical flat glass plate in a holder. The contact between the two surfaces can be made watertight by mechanical pressing alone because of their optical flatness. The microgrooves in the silicon microchannel chip were prefilled with saline.

Within 10 min of collecting blood into heparinized tubes, 0.1 mL of blood was drawn through the microchannels as an ex vivo capillary model (7854-parallel, 7 × 4.5-μm equivalent cross-section, 30 µm long) under a constant vacuum of 20 cm H_2_O (1.96 kPa). The time required for 0.1 mL of saline to pass through the microchannels was determined for calibration before each blood measurement. Microscopic motion images of blood passing through the microchannels were monitored and stored on a computer. Once 0.08–0.10 mL of blood had exited the microchannel array, five fields were recorded, five still images were randomly selected for off-line analysis, and the number of adhesive or clumped leukocytes on the microchannel platforms in these images were counted. Adhesive leukocytes were defined as static leukocytes with clear surface borders on the still images.

### 2.3. Measurement of Derivatives of Reactive Oxygen Metabolites in Serum

We measured hydroperoxide levels as serum levels of diacron reactive oxygen metabolite derivatives (d-ROMs) using a FREE Carpe Diem photometer (Diacron srl, Grosseto, Italy). The d-ROM test depends on a Fenton-like reaction to produce lipid peroxy and alkoxy radicals, which in turn react with chromogenic substrates.

### 2.4. Statistical Analysis

All statistical analyses were performed using EZR (Saitama Medical Center, Jichi Medical University, Saitama, Japan), a modified graphical user interface for R (The R Foundation for Statistical Computing, Vienna, Austria) designed to add statistical functions frequently used in biostatistics. Data are presented as the mean ± standard deviation (SD) for continuous variables unless otherwise indicated, and as numbers and percentages for categorical variables. Baseline comparisons were conducted using the Wilcoxon rank sum test, Student’s *t*-test, chi-squared test, and Fisher’s exact test. Comparisons of the time-response curves of various parameters, such as blood glucose, total cholesterol, LDL-C, TG, HDL-C, remnant lipoprotein cholesterol (RLP-C), biological antioxidant potential (BAP), d-ROMs, FFA, whole blood transit time, and the number of adhered leukocytes, of the two groups were made using two-way repeated measures analysis of variance (ANOVA). When *p* was <0.05 for the ANOVA analyses, these were followed by the Tukey–Kramer post hoc test. Subgroup analyses were conducted for serial changes in plasma FFA values in the group where the baseline plasma FFA level was above the median, as well as for serial changes in whole blood transit time in the group where the baseline whole blood transit time was above the median. *p*-values of <0.05 were considered statistically significant.

Sample size calculation was performed using G*Power (Heinrich-Heine-University Düsseldorf, North Rhine-Westphalia, Germany). The number of study participants was set based on studies in which whole blood transit time was measured in men before and after FFA provocation by lipid/heparin infusion [15,16,17]. The target number of patients was 41 for each group, totaling 82 (α = 0.05, 1 − β = 0.80). We set the dropout rate for this study at 10%, and the required sample size was therefore 92 patients.

## 3. Results

Ninety-six patients, of which fifty patients received 0.2 mg/day of pemafibrate and forty-six patients did not receive pemafibrate or other fibrates, participated in this study for 16 weeks. The clinical characteristics of the two groups are summarized in Table 1. All patients had similar baseline characteristics. Bodyweight, heart rate, and hematocrit after pemafibrate administration did not differ across experimental days. No significant differences in the glycemic parameters between the groups were observed (Figure 1a). The pemafibrate group showed a slight decrease from the baseline HbA1c level at week 16, although this was not statistically significant when compared with the non-pemafibrate group. Fasting serum TG levels of the non-pemafibrate group decreased from a baseline of 245.8 ± 112.6 mg/dL to 205 ± 139.6 mg/dL at week 8, and 220.5 ± 148.6 mg/dL at week 16 (Figure 1d). The pemafibrate group fasting serum TG levels decreased from a baseline of 278.6 ± 122.2 mg/dL to 161.8 ± 101.1 mg/dL at week 8, and 135.5 ± 49.1 mg/dL at week 16 (Figure 1d). RLP-C levels in the pemafibrate group decreased significantly from a baseline of 13.2 ± 8.7 mg/dL to 5.5 ± 3.1 mg/dL at week 8, and 4.9 ± 2.5 mg/dL at week 16 (Figure 1f). Total cholesterol, LDL-C, HDL-C, and FFA levels remained unchanged in both groups (Figure 1b,c,e,i). Regarding liver enzymes, the aspartate oxoglutarate aminotransferase (ALT) and γ-glutamyl transferase (GTP) levels decreased, and the alanine oxoglutarate aminotransferase (AST) and creatine phosphokinase (CK) levels were not significantly altered with pemafibrate treatment (Appendix A).

### 3.1. Effect of Reduced TG Levels on Hemorheology in Ex Vivo Microvascular Models

Figure 1j shows serial changes in whole blood transit time, reflecting the apparent relative viscosity of whole blood in the MCFAN. There was no difference between the two groups after two-way repeated ANOVA. The whole blood transit time 16 weeks after the start of pemafibrate administration tended to be decreased compared with the baseline value. Subgroup analysis for whole blood transit time values for the group above the median at baseline (Figure 2a), showed no significant difference between the two groups. Comparison of the adherent leukocyte counts also showed no significant difference between the two groups (Figure 1k).

### 3.2. Effect of Pemafibrate on Biological Antioxidant Potential and Oxidative stress

Pemafibrate initiation time-dependently and significantly improved the BAP compared with the non-pemafibrate group (*p* = 0.041, Figure 1g). However, no significant change in hydroperoxide serum levels was measured using the d-ROM test in either of the groups (Figure 1h).

### 3.3. Effect of Pemafibrate on FFA Levels

There was no difference in overall serum FFA levels throughout the study period between the groups (Figure 1i). In a subgroup analysis with serum FFA levels above the median at baseline, there was no significant interaction (Figure 2b), however the pemafibrate group showed the significant suppression of plasma FFA levels at week 16 compared with the baseline after one-way ANOVA (*p* = 0.034).

## 4. Discussion

In this nonrandomized, controlled trial of patients with type 2 diabetes mellitus or metabolic syndrome complicated by hypertriglyceridemia and with a prolonged whole blood transit time on MCFAN, we found that 16 weeks of pemafibrate treatment did not improve blood rheology (whole blood transit time or leukocyte activation in MCFAN) or serum FFA. Pemafibrate treatment reduced fasting serum TG levels by an average of 38.6% at week 8, and this significant reduction in TG levels remained stable for 16 weeks. In addition to TG, RLP-C, a marker of lipoproteins, was reduced by an average of 50.7%. To the best of our knowledge, this is the first study to investigate the effects of pemafibrate on blood rheology in patients with type 2 diabetes mellitus or metabolic syndrome complicated by hypertriglyceridemia.

Our results of lipid profile changes by pemafibrate correspond with the most recent report of the PROMINENT clinical trial, a double-blind, randomized, placebo-controlled trial of pemafibrate [23], which found that, although TG, very low density lipoprotein cholesterol (VLDL-C), RLP-C, and apolipoprotein C-III levels were reduced in the pemafibrate group, the rate of cardiovascular events was not different from that of the placebo group [23]. These pemafibrate-mediated decreases in triglyceride rich lipoprotein (TRL), without changes in non-HDL cholesterol and total cholesterol levels, suggest that atherogenic TRL and their RLP-C are not reduced and may retain their elevated plasma concentrations without improvement in heterogeneity. In these conditions, the inflammatory response creates the possibility of endothelial dysfunction [24,25], which may inhibit the atheroprotective and anti-inflammatory effects of HDL-C [26,27]. The PPAR is a type of nuclear receptor and a transcription factor that transmits signals from lipophilic factors to the genome, and the selective peroxisome proliferator-activated receptor modulatorα (SPPARMα) increases the transcriptional activity of PPARα. Since SPPARMα binds directly to DNA promoters and inhibits their transcription, the possibility that pemafibrate-mediated changes in lipid metabolism do not directly reduce TG alone, but are accompanied by increased HDL-C and LDL-C levels (thus the balance of lipid metabolism and the improvement in LDL-C risk are not substantially altered), should be considered in the next research phase [28,29,30]. Pemafibrate is a potent and selective synthetic agonist of the nuclear receptor PPARα, which increases the activity of lipoprotein lipase, an enzyme essential for the hydrolysis of VLDL-C and chylomicron triglycerides, and decreases TG levels by lowering hepatic VLDL-C apolipoprotein B levels and TG production [31,32,33,34,35,36]. Medium-sized TRL particles have been demonstrated to enter the arterial lumen at a reduced rate compared with that of smaller LDL particles [37,38]. Due to their larger molecular size, triglyceride-rich lipoproteins entering the intima have greater difficulty moving against the pressure gradient than LDL particles, and are therefore preferentially trapped by arterial luminal components [36,37,38,39]. In fact, the pemafibrate-mediated reduction of TG and RLP-C is accompanied by slightly increased LDL-C and HDL-C with no overall change in total cholesterol levels, which is similar to the results of the PROMINENT clinical trial [23].

Since only 30% of patients in the pemafibrate group had elevated baseline serum FFA levels, a subgroup analysis was conducted for the group with above-median serum FFA levels. Although pemafibrate reduced the serum FFA levels of this subgroup by 34.6% over 16 weeks, the reduction in FFA levels did not correlate with leukocyte activation or a reduction in whole blood transit time. There were also no significant differences in leukocyte activation or whole blood transit time between the group with baseline FFA levels > the median. We have previously reported that the renin–angiotensin system in leukocytes plays a pivotal role in the development of endothelial dysfunction with high FFA levels [16,17,18]. Since excess FFA from visceral adipose tissue induces abnormal vascular responses and insulin resistance by inducing oxidative stress and inflammatory responses [40], we hypothesized that lowering FFA levels with pemafibrate would prevent endothelial dysfunction and normalize blood rheology. However, our results showed no clearly significant effects of pemafibrate on hemorheological data in patients with concurrently elevated TG levels and type 2 diabetes or metabolic syndrome. The lack of benefit from pemafibrate was thought to be due to the small percentage of patients with high FFA. A possible reason for this is that we did not add FFA values to the entry criteria, as FFA values are not routinely measured in our clinical practice and were difficult to use for screening. Since RLP promotes platelet aggregation and increases blood viscosity, a decrease in RLP with pemafibrate may lead to a reduction in whole blood transit time; however, in this study, no reduction in whole blood transit time was observed and no change in leukocyte adhesive capacity was observed. The reason for this is unknown. It has been suggested that pemafibrate may indirectly stabilize the dynamic equilibrium of fatty acids by converting many TRL residues to LDL-C, thereby lowering elevated serum FFA levels to the normal range [41]. Abdominal obesity, as seen in hypertriglyceridemia and metabolic syndrome, increases total body fat and releases large amounts of fatty acids owing to the increased adipocyte mass [42]. However, although obesity worsens the general health status, excess fat does not necessarily cause metabolic abnormalities, and the loss of the dynamic equilibrium between fatty acid release and consumption and the increase in serum FFA levels are thought to be responsible for endothelial dysfunction within blood vessels [43]. 

This study has some limitations. First, this study was a single-center, nonrandomized, controlled study with a limited sample size. The statin therapy was significantly lower in the pemafibrate group compared with the non-pemafibrate group. Second, only 30% of the study patients had elevated baseline serum FFA levels; therefore, further research is needed in patients with concurrently elevated FFA and TG levels and type 2 diabetes.

## 5. Conclusions

In conclusion, although pemafibrate treatment decreased serum levels of TG, VLDL-C, and RLP-C, it did not significantly improve blood rheology as reflected by whole blood transit time and leukocyte activity in patients with type 2 diabetes mellitus or metabolic syndrome complicated by hypertriglyceridemia.

## Figures and Tables

**Figure 1 jcm-12-01481-f001:**
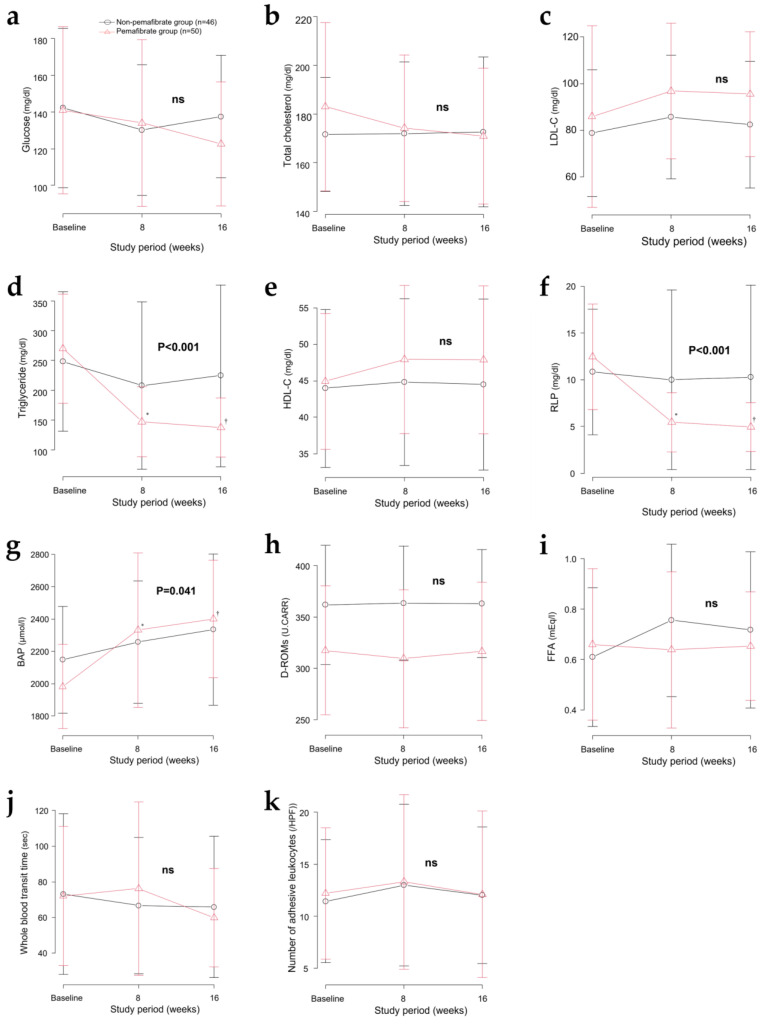
Lipid levels and related variables over 16 weeks. Values are presented as mean ± SD. Serial changes in (**a**) blood glucose, (**b**) total cholesterol, (**c**) low-density lipoprotein cholesterol (LDL-C), (**d**) triglyceride, (**e**) high-density lipoprotein cholesterol (HDL-C), (**f**) remnant-like particle cholesterol (RLP-C), (**g**) biological antioxidant potential (BAP), (**h**) diacron reactive oxygen metabolites (dROMs), (**i**) free fatty acid (FFA), (**j**) whole blood transit time, (**k**) number of adhesive leukocytes were presented. * *p* < 0.05 vs. baseline for 0.2 mg/d pemafibrate at week 8. † *p* < 0.05 vs. baseline for 0.2 mg/day pemafibrate at week 16.

**Figure 2 jcm-12-01481-f002:**
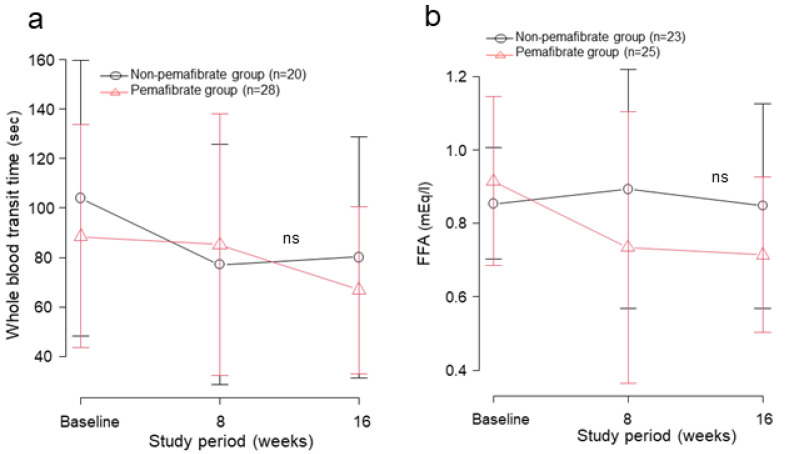
Subgroup analysis for (**a**) whole blood transit times and (**b**) serum free fatty acid (FFA) above the median at baseline. No significant difference between the two groups is seen for serial changes in (**a**) whole blood transit times or (**b**) serum FFA values in subgroups with baseline values above the median. Pemafibrate tended to decrease serum FFA levels both at 8 and 16 weeks. Values are presented as mean ± SD.

**Table 1 jcm-12-01481-t001:** Patient characteristics. Data are presented as mean ± SD for continuous parameters and % (*n*) for categorical parameters.

	Overall (*n* = 96)	Non–Pemafibrate (*n* = 46)	Pemafibrate (*n* = 50)	*p* Value
Age (years)_(IQR)_	67.39 (61–74)	67.2 (60.2–74)	67.5 (62–73.7)	0.9
Female	79.1% (20)	21.7% (10)	20% (10)	1
Hight (cm)	165.0 ± 7.7	165.5 ± 7.8	164.5 ± 7.6	0.56
Bodyweight (kg)	71.8 ± 11.0	73.3 ± 12.4	70.6 ± 9.8	0.25
Smoking habit	30.2% (29)	28.2% (13)	32% (16)	0.58
Drinking habit	47.9% (46)	47.8% (22)	48% (24)	1
Diabetes mellitus	79.1% (71)	76% (35)	72% (36)	0.82
Duration of diabetes (years)	11.5 ± 8.2	12.8 ± 8.4	10.1 ± 8.0	0.18
Diabetic retinopathy	11.4% (11)	13% (6)	10% (5)	<0.05
Old myocardial infarction	7.2% (7)	8.6% (4)	6% (3)	0.71
Angina pectoris	9.3% (9)	8.6% (4)	10% (5)	1
Percutaneous coronary intervention	7.2% (7)	2.1% (1)	12% (6)	0.11
Peripheral artery disease	2% (2)	2.1% (1)	2% (1)	1
Congestive heart failure	5.2% (5)	8.6% (4)	2% (1)	0.19
Arrhythmia	9.3% (9)	13% (6)	6% (3)	0.3
Hypertension	64.5% (62)	67.3% (31)	62% (31)	0.67
Chronic kidney disease	28.7% (19)	30.4% (14)	10% (5)	<0.05
Liver disfunction	17.7% (17)	23.9% (11)	32% (16)	0.5
DPP–4	37.5% (36)	43.4% (20)	32% (16)	0.29
Statin	56.2% (54)	67.3% (31)	46% (23)	<0.05
Leukocyte (×10^3^/μL)	7.1 ± 1.8	7.2 ± 1.7	7.0 ± 2.0	0.5
Hematocrit (%)	44.1 ± 3.4	43.4 ± 3.4	44.8 ± 3.0	<0.05
Platelet (×10^4^/μL)	220.1 ± 57.4	223.4 ± 65.3	217.1 ± 50.3	0.60
Glucose (mg/dL)	140.3 ± 43.7	139.1 ± 42.2	141 ± 45.8	0.80
HbA1c (%)	7.12 ± 1.21	7.2 ± 1.17	7.0 ± 1.26	0.59
CPK (mg/dL)	108.9 ± 59.7	118.08 ± 69.2	100.4 ± 49.2	0.15
AST (mg/dL)	29.7 ± 15.4	27.4 ± 11.8	31.7 ± 18.1	0.18
ALT (mg/dL)	28.5 ± 16.4	28.6 ± 16.4	28.4 ± 16.8	0.95
γGTP (mg/dL)	59.8 ± 80.0	49.6 ± 48.2	69.4 ± 101.4	0.23
Total cholesterol (mg/dL)	178.5 ± 31.3	171.7 ± 23.4	184.8 ± 36.6	<0.05
Low–Density Lipoprotein cholesterol (mg/dL)	82.4 ± 33.3	79.4 ± 27.3	85.1 ± 38.5	0.41
Triglyceride (mg/dL)	262.9 ± 117.6	245.8 ± 112.6	278.6 ± 122.2	0.18
High–Density Lipoprotein cholesterol (mg/dL)	44.7 ± 9.9	44.1 ± 10.5	45.2 ± 9.4	0.60
Remnant like particles cholesterol (mg/dL)	11.9 ± 7.7	10.6 ± 6.4	13.2 ± 8.7	0.10
hsCRP (mg/dL)	0.13 ± 0.15	0.15 ± 0.14	0.12 ± 0.16	0.30
BAP (μmol/L)	2078.1 ± 304.9	2164.7 ± 331.7	1998.4 ± 260.0	<0.01
d–ROMs (U.CARR)	338.0 ± 66.2	358.3 ± 65.7	319.4 ± 62.3	<0.01
FFA (mEq/L)	0.66 ± 0.31	0.63 ± 0.29	0.69 ± 0.32	0.32
Whole blood transit time (s)	71.2 ± 40.3	71.6 ± 43.3	70.9 ± 38.1	0.93
Number of adhesive leukocytes (/HPF)	11.7 ± 5.7	11.6 ± 5.7	11.9 ± 5.8	0.79

Abbreviations: DPP-4, dipeptidyl peptidase-4; CPK, creatin phosphokinase; AST, aspartate aminotransferase; ALT, alanine aminotransferase; GTP, γ-glutamyl transpeptidase; RLP-C, remnant-like particle cholesterol; hsCRP, high sensitivity C-reactive protein; BAP, biological antioxidant potential; d-ROMs, diacron reactive oxygen metabolites; FFA, free fatty acid.

## Data Availability

Not applicable.

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
