# Peer review of "Effect of Pemafibrate on Hemorheology in Patients with Hypertriglyceridemia and Aggravated Blood Fluidity Associated with Type 2 Diabetes or Metabolic Syndrome"

_jcm, 2023, doi:10.3390/jcm12041481_

Round 1
Reviewer 1 Report
This manuscript studies the effects of Pemafibrate on Hemorheology in Patients with Hypertriglyceridemia and Type 2 Diabetes and/or Metabolic Syndrome. It is short, well written in all its sections. A few comments that could help to improve the manuscript:
1. Authors could elaborate more about how they decided to exclude patients deemed by the principal investigator to be inappropriate for participation in the study.
2. The method to assess “Whole blood Rheology” can be further explained. It is essential for this research and authors can provide more information to readers that are les knowledgeable on how to carry out these studies.
3. The writing of lines 200-204 and 209 and 214 in the Discussion could be reviewed and clarified.
Author Response
Response to Comments by Reviewer 1
Thank you for carefully reviewing our manuscript entitled “Long-term tailor-made exercise intervention reduces the risk of developing cardiovascular diseases and all-cause mortality in patients with diabetic kidney disease” (manuscript ID: jcm- 2077140).
Your comments were very helpful in improving our manuscript and have provided essential guidance for our research. We hope that our responses and the revised version of our manuscript adequately address your concerns.
Comment 1:
This manuscript studies the effects of Pemafibrate on Hemorheology in Patients with Hypertriglyceridemia and Type 2 Diabetes and/or Metabolic Syndrome. It is short, well written in all its sections. A few comments that could help to improve the manuscript:
- Authors could elaborate more about how they decided to exclude patients deemed by the principal investigator to be inappropriate for participation in the study.
Authors’ Response 1:
Thank you for pointing this out. We have expanded the section according to your suggestion as follows:
“The exclusion criteria were as follows: 1) patient treated by pemafibrate, fibrates, or omega-3 fatty acids; 2) patients deemed by the principal investigator or sub-investigator to be inappropriate for participation in the study, such as those with poor medication adherence, dementia (Hasegawa’s dementia scale <20/30), or alcoholism.
Comment 2:
The method to assess “Whole blood Rheology” can be further explained. It is essential for this research and authors can provide more information to readers that are less knowledgeable on how to carry out these studies.
Authors’ Response 2:
Thank you for your suggestion. We have added two more references (20,21) and a more detailed explanation regarding the methodology of the microchannel flow analyzer.
Page 2, lines 77–83. “We used a microchannel flow analyzer as an ex vivo model of capillaries and arterioles to assess whole blood rheology and leukocyte activity as previously described [16,17, 20, 21]. Briefly, microgrooves that formed on the surface of a silicon chip were converted to leak-proof microchannels by being tightly covered with an optical flat glass plate in a holder. The contact between the two surfaces can be made watertight by mechanical pressing alone because of their optical flatness. The microgrooves in the silicon microchannel chip were prefilled with saline.
- Kikuchi Y, Sato K, Ohki H, Kaneko T. Optically accessible microchannels formed in a single-crystal silicon substrate for studies of blood rheology. Microvasc Res 1992, 44, 226–240.
- Kikuchi Y, Sato K, Mizuguchi Y. Modified cell-flow microchannels in a single-crystal silicon substrate and flow behavior of blood cells. Microvasc Res 1994, 47, 126–139.
Comment 3:
The writing of lines 200-204 and 209 and 214 in the Discussion could be reviewed and clarified.
Authors’ Response 3:
We thank you for this suggestion. Accordingly, we have revised this part for clarity as follows:
Lines 200–204 (corresponding lines in revised manuscript: ):
Since SPPARMα binds directly to DNA promoters and inhibits their transcription, the possibility that pemafibrate-mediated changes in lipid metabolism do not directly reduce TG alone, but are accompanied by increased HDL-C and LDL-C levels (thus the balance of lipid metabolism and improvement of LDL-C risk are not substantially altered), should be considered in the next research phase [26–28].
Line 209 (corresponding page and line in revised manuscript: ):
Due to their larger molecular size, triglyceride-rich lipoproteins entering the intima have greater difficulty moving against the pressure gradient than LDL particles, and are therefore preferentially trapped by arterial luminal components [36–38].
Line 214: (corresponding page and line in revised manuscript: ):
Indeed, as shown in the PROMINENT clinical trial [21], pemafibrate-mediated reductions in TG and RLP-C are accompanied by slightly elevated LDL-C and HDL-C with no overall change in total cholesterol levels.
Reviewer 2 Report
Review for Manuscript Number jcm-2077140
Dear Authors,
I highly appreciate your hard work on this paper and congratulations for the very up-to-date methodology. I would like to provide some suggestions and questions for improving the manuscript:
1. How did you check normal distribution before you have performed your parametric tests? Which test did you use? Which parameters were normal distributed?
2. The statin therapy was significantly lower in the pemafibrate group; could it have a role in your results?
3. Patients with diabetes should reach an LDL cholesterol treatment goal <100mg/dL. The mean in case of your patients is under this level, but how many of your patients reached this treatment goal in real?
Hope you succeed in publishing your work.
With best regards,
Author Response
Response to Comments by Reviewer 2
Thank you for carefully reviewing our manuscript entitled “Long-term tailor-made exercise intervention reduces the risk of developing cardiovascular diseases and all-cause mortality in patients with diabetic kidney disease” (manuscript ID: jcm- 2077140).
Your comments were very helpful in improving our manuscript and have provided essential guidance for our research. We hope that our responses and the revised version of our manuscript adequately address your concerns.
Comment 1:
How did you check normal distribution before you have performed your parametric tests? Which test did you use? Which parameters were normal distributed?
Author’s Response 1:
We thank you for your suggestion. We have checked whether the variables, in particular the primary outcome (whole blood transit time as a rheological parameter), were normally distributed by assessing the density curb and qq-plot.
Below are the plots of the outcome.
Chart 1: Outcome at baseline: Please check our figures in PDF
Chart 2: QQ plot at baseline
Even though the distribution was slightly skewed due to outliers, the density curb did not significantly violate the concept of normal distribution.
Chart 3: Outcome at 8 weeks
Chart 4: QQ plot at 8 weeks
Chart 5: Outcome at 16 weeks
Chart 6: Outcome at 16 weeks
Interestingly, the more we followed the outcome, the more normally distributed the outcome was becoming. Ideally, we should have used the Shapiro–Wilk test, however, our real-world dataset was not large and had a few outlines; therefore, according to our biostatistician, a density curb was more reliable. For these reasons, we considered the outcome to be normally distributed.
Comment 2:
The statin therapy was significantly lower in the pemafibrate group; could it have a role in your results?
Author’s Response 2:
Yes, you are correct. This is a limitation of our non-randomized control study. We have added this issue to the limitations as follows:
Page 8, lines 252–253: The statin therapy was significantly lower in the pemafibrate group compared with the non-pemafibrate group.
Comment 2:
Patients with diabetes should reach an LDL cholesterol treatment goal <100mg/dL. The mean in case of your patients is under this level, but how many of your patients reached this treatment goal in real?
Author’s Response 2:
According to the 2022 Japanese Guidelines for Prevention of Atherosclerotic Diseases for Management of Dyslipidemia, the LDL cholesterol treatment goal is <120 mg/dL.
At the start of the study, 72 patients had LDL cholesterol levels of <100 mg/dL; after 16 weeks, 73 patients had achieved levels of <100 mg/dL.